# Association of subcutaneous and visceral adipose tissue with overall survival in Taiwanese patients with bone metastases – results from a retrospective analysis of consecutively collected data

Wen Ching Chuang [1,2ᴑ], Ngan Ming Tsang[3,4ᴑ]*, Chi Cheng Chuang[5], Kai Ping Chang[6], Ping Ching Pai[3], Kuan Hung Chen[1,2], Wen Chi Chou[7], Shiao Fwu Tai[8], Shu Chen Liu[9], Kin Fong Lei[10]

1 Chang Gung University, Medicine, Taoyuan, Taiwan, 2 Chang Gung Memorial Hospital, Linkou Branch, Taoyuan, Taiwan, 3 Department of Radiation Oncology, Linkou Chang Gung Memorial Hospital and Chang Gung University, Taoyuan, Taiwan, 4 School of Traditional Chinese Medicine, Chang Gung University, Taoyuan, Taiwan, 5 Department of Neurosurgery, Chang Gung Memorial Hospital and University at Lin-Kou, Taoyuan, Taiwan, 6 Department of Otolaryngology-Head Neck Surgery, Linkou Chang Gung Memorial Hospital and Chang Gung University at Lin-Kou, Taoyuan, Taiwan, 7 Division of Hematology-Oncology, Department of Internal Medicine, Chang Gung Memorial Hospital Linkou Branch, and School of Medicine, Chang Gung, Taoyuan, Taiwan, 8 Department of Otorhinolaryngology, Chang Gung Memorial Hospital, Linkou Branch, Taoyuan, Taiwan, 9 Department of Biomedical Sciences and Engineering, National Central University, Taoyuan, Taiwan, 10 Graduate Institute of Biomedical Engineering, Chang Gung University, Taoyuan, Taiwan

ᴑ These authors contributed equally to this work.
* vstsang@cgmh.org.tw

**Data Availability Statement:** Data cannot be shared publicly because of patient privacy. Data are

## Abstract

### Background

Growing evidence indicates that measures of body composition may be related to clinical outcomes in patients with malignancies. The aim of this study was to investigate whether measures of regional adiposity–including subcutaneous adipose tissue index (SATI) and visceral adipose tissue index (VATI)–can be associated with overall survival (OS) in Taiwanese patients with bone metastases.

### Methods

This is a retrospective analysis of prospectively collected data. We examined 1280 patients with bone metastases who had undergone radiotherapy (RT) between March 2005 and August 2013. Body composition (SATI, VATI, and muscle index) was assessed by computed tomography at the third lumbar vertebra and normalized for patient height. Patients were divided into low- and high-adiposity groups (for both SATI and VATI) according to sex-specific median values.

### Results

Both SATI (hazard ratio [HR]: 0.696; P<0.001) and VATI (HR: 0.87; P = 0.037)–but not muscle index–were independently associated with a more favorable OS, with the former

available from the the Institution Review Board of the Chang Gung Memorial Hospital (approval number: IRB: 201701224B0,contact via irb1@cgmh.org.tw) for researchers who meet the criteria for access to confidential data.

**Funding:** This work was supported by Ministry of Science and Technology, Taiwan [NZRPD1H1231] and Chang Gung Memorial Hospital (CGMH) research grants [CMRPG3H1221, CMRPG3H1851, and BMRP238]. The funders had no role in study design, data collection and analysis, decision to publish, or preparation of the manuscript.

**Competing interests:** The authors have declared that no competing interests exist.

showing a stronger relationship. The most favorable OS was observed in women with high SATI (11.21 months; 95% confidence interval: 9.434−12.988; P<0.001).

## Conclusions

High SATI and VATI are associated with a more favorable OS in Taiwanese patients with bone metastases referred for RT. The question as to whether clinical measures aimed at improving adiposity may improve OS in this clinical population deserves further scrutiny.

## Introduction

Bone represents one of the most common sites for cancer spread, especially in patients with breast, prostate, or lung malignancies.[1, 2] Bone metastases are a significant source of morbidity, decreased performance status, and impaired quality of life. Moreover, the presence of bone metastases typically portends a poor prognosis, with a median overall survival (OS) of 6–7 months.[3]

Several factors–including clinical stage, patient demographics, and tumor histology–have been shown to affect the OS of patients with bone metastases.[4] Notably, sex disparities have been reported in the survival of patients with metastatic spread to the bone–with mortality rate ratios being significantly higher in males than in females for most malignancies.[5] Women also have a higher total adiposity than men, with a preponderance of subcutaneous adipose tissue. In contrast, men typically tend to accumulate visceral adipose tissue.[6] Subcutaneous and visceral adipose tissue indices (SATI and VATI, respectively) may influence the clinical outcomes of patients with cancer in a sex-dependent manner. [7]

Previous studies have reported a significant prognostic impact of SATI and VATI in different solid tumors, including advanced renal cell carcinoma, hepatocellular carcinoma, and pancreatic cancer,[8–10] although there has been some discrepant findings and the therapeutic implications of these observations have not been fully elucidated [11].

The aim of this study was to investigate whether measures of regional adiposity–including SATI and VATI–can be associated with overall survival (OS) in Taiwanese patients with bone metastases who were referred for radiotherapy (RT).

## Materials and methods

### Study patients

The present study was designed as a retrospective review of prospectively collected data and was conducted in a radiation oncology setting. Between March 2005 and August 2013, a total of 1654 Taiwanese patients with bone metastases were consecutively referred for RT to the Chang Gung Memorial Hospital. All of them had a histology-proven diagnosis of cancer and underwent computed tomography (CT) imaging within 30 days of the initial assessment. The diagnosis of bone metastases was based on the results of bone scintigraphy, X-ray, CT, or magnetic resonance imaging. Patients were excluded in presence of the following criteria: age <18 years, unavailable CT scans within two weeks before the start of RT, or lack of measures of weight and height within two weeks of enrolment. A total of 374 cases met the exclusion criteria, resulting in a final study sample of 1280 patients. The study protocol was reviewed and approved by the Institution Review Board of the Chang Gung Memorial Hospital (approval number: IRB: 201701224B0). Owing to the retrospective nature of the analysis, the need for informed consent was waived. Data collection from electronic medical records was supervised by an experienced nurse and a radiation oncologist.

## CT-based body composition analysis

In keeping with previous methodology,[12] single-slice CT imaging at level L3 was used to analyze adiposity. SATI and VATI were identified according to Hounsfield units (HU) (from -190 to -30 HU for SATI and from -29 to 150 HU for VATI, respectively). The tissue cross-sectional areas (expressed in $cm^2$) were calculated automatically by the CT software after normalization for patient height. SATI, VATI, total adipose tissue, and skeletal muscle indexes were expressed in $cm^2 \, m^{-2}$. All adiposity measures were taken in the two weeks preceding the start of RT.

## Variable definition

Owing to the lack of a commonly accepted standard, SATI, VATI, and skeletal muscle indices were dichotomized according to median values measured at L3. OS was defined as the time elapsed from the start of RT for bone metastases to the date of death. Body mass index (BMI) was categorized as follows: underweight (BMI <18.5 $kg/m^2$), normal weight (BMI: 18.5–24.99 $kg/m^2$), overweight (BMI: 25–29.99 $kg/m^2$), and obese (BMI $\geq$30 $kg/m^2$). Equivalent doses in 2-Gy fraction (EQD2Gy) were used to express different total radiation doses in terms of amount and number of fractions. The time to metastases (calculated from the time of diagnosis of primary cancer to the identification of distant metastases) was categorized in $\leq$ 1 year *versus* >1 year. Metastases were considered multiple in presence of simultaneous involvement of at least two organs or different parts of skeleton (e.g., sternum and sacrum). The use of systemic therapy was investigated in the timeframe ranging from 1 month before RT to the date of the last follow-up. Other variables of interest were previously described.[13] The presence of comorbidities was dichotomized (yes *versus* no) according to the Charlson comorbidity index. Employment status was classified into three categories using the Registrar General's Social Class (RGSC) scheme, as follows: unemployed, low-wage employed, and high-wage employed. Education status was categorized as high *versus* low (junior high school and above *versus* elementary school and below). The patient's place of residence was dichotomized as either rural or urban (population density below or above 800 persons per $km^2$, respectively). Risky oral habits were classified as follows: cigarette smoking (smoked $\geq$100 in lifetime *versus* < 100 cigarettes in lifetime and no current smoking), betel quid chewing (current/former *versus* never), and alcohol drinking (current/former *versus* never).

## Statistical analysis

Continuous variables were compared using the Student's *t*-test, whereas the Pearson's chi-square test was used for categorical variables. The associations between the study variables (including indices of adiposity) and OS were investigated using univariate and multivariate Cox proportional hazard ratio analyses. Results were expressed as hazard ratios (HRs) with their 95% confidence interval (CIs). We also categorized patients according to SATI and VATI values (high *versus* low, with high values serving as references). Survival plots were constructed with the Kaplan-Meier method (log-rank test).Two-tailed P values <0.05 were considered statistically significant. Owing to the exploratory nature of the study, the Bonferroni's correction was not applied.

# Results

## Patient characteristics

The general characteristics of the study patients are summarized in Table 1. Of the 1280 participants, 1237 were followed up until death, whereas the remaining 43 were censored on the

**Table 1. Patient characteristics according to the subcutaneous and visceral adiposity status.**

| | | SATI | | | | VATI | | | |
|---|---|---|---|---|---|---|---|---|---|
| | | Low | High | Total number | P value | Low | High | Total number | P value |
| **Number of patients** | | 640 (50.0%) | 640 (50.0%) | 1280 (100%) | | 640 (50.0%) | 640 (50.0%) | 1280 (100%) | |
| **SATI** | | | | | | | | | |
| | Low | | | | | 455 (71.1%) | 185 (28.9%) | 640 (50.0%) | <0.001[a] |
| | High | | | | | 185 (28.9%) | 455 (71.1%) | 640 (50.0%) | |
| | Median | 8.15 (0.01–15.48) | 27.77 (15.50–148.77) | 15.49 (0.01–148.77) | <0.001[b] | 9.93 (0.01–71.92) | 23.30 (1.24–148.77) | 15.49 (0.01–148.77) | <0.001[b] |
| | Mean ± SD, cm²/m² | 7.96±4.58 | 33.42±17.87 | 20.69±18.22 | | 12.81±11.92 | 28.57±19.96 | 20.69±18.22 | |
| **VATI** | | | | | | | | | |
| | Low | 455 (71.1%) | 185 (28.9%) | 640 (50.0%) | <0.001[a] | | | | |
| | High | 185 (28.9%) | 455 (71.1%) | 640 (50.05%) | | | | | |
| | Median | 4.74 (0.02–72.53) | 15.59 (0.40–93.34) | 9.84 (0.02–93.34) | <0.001[b] | 3.70 (0.02–9.83) | 19.35 (9.84–93.34) | 9.84 (0.02–93.34) | <0.001[b] |
| | Mean | 8.10±9.39 | 18.35±13.19 | 13.23±12.54 | | 4.07±2.92 | 22.39±11.75 | 13.23±12.54 | |
| **Muscle index** | | | | | | | | | |
| | Low | 293 (45.8%) | 347 (54.2%) | 640 (50.0%) | 0.003[a] | 333 (52.0%) | 307 (48.0%) | 640 (50.0%) | 0.162[a] |
| | High | 347 (54.2%) | 293 (45.8%) | 640 (50.0%) | | 307 (48.0%) | 333 (52.0%) | 640 (50.0%) | |
| | Median | 17.23 (4.02–49.49) | 16.01 (6.44–93.90) | 16.61 (4.02–93.90) | 0.015[b] | 16.41 (4.02–43.93) | 16.89 (6.44–93.90) | 16.61 (4.02–93.9) | 0.063[b] |
| | Mean | 17.78±5.59 | 16.96±6.28 | 17.37±5.95 | | 17.06±5.65 | 17.68±6.23 | 17.37±5.95 | |
| **Age group, years** | | | | | | | | | |
| | <59.5 | 318 (49.7%) | 313 (48.9%) | 631 (49.3%) | 0.823[a] | 376 (58.8%) | 255 (39.8%) | 631 (49.3%) | <0.001[a] |
| | ≥59.5 | 322 (50.3%) | 327 (51.1%) | 649 (50.7%) | | 264 (41.3%) | 385 (60.2%) | 649 (50.7%) | |
| | Median | 59.67 (19.54–95.57) | 59.83 (21.98–87.05) | 59.73 (19.54–95.57) | 0.503[b] | 56. 79 (19.54–95.57) | 63.01 (27.9–90.45) | 59.73 (19.54–95.57) | <0.001[b] |
| | Mean | 60.55±13.16 | 60.08±12.11 | 60.32±12.64 | | 57.59±12.94 | 63.04±11.73 | 60.32±12.64 | |
| **Sex** | | | | | | | | | |
| | Female | 152 (23.8%) | 388 (60.6%) | 540 (42.2%) | <0.001[a] | 287 (44.8%) | 253 (39.5%) | 540 (42.2%) | 0.062[a] |
| | Male | 488 (76.3%) | 252 (39.4%) | 740 (57.8%) | | 353 (55.2%) | 387 (60.5%) | 740 (57.8%) | |
| **Performance status** | | | | | | | | | |
| | ECOG 0–1 | 428 (66.9%) | 469 (73.3%) | 897 (70.1%) | 0.015[a] | 444 (69.4%) | 453 (70.8%) | 897 (70.1%) | 0.625[a] |
| | ECOG 2–4 | 212 (33.1%) | 171 (26.7%) | 383 (29.9%) | | 196 (30.6%) | 187 (29.2%) | 383 (29.9%) | |
| **Onset of metastasis** | | | | | | | | | |
| | ≤ 1 year | 470 (73.4%) | 430 (67.2%) | 900 (70.3%) | 0.017[a] | 455 (71.1%) | 445 (69.5%) | 900 (70.3%) | 0.541[a] |
| | > 1 years | 170 (26.6%) | 210 (32.8%) | 380 (29.7%) | | 185 (28.9%) | 195 (30/5%) | 380 (29.7%) | |
| | Median | 0.08 (0–13.50) | 0.04 (0–15.64) | 0.05 (0–15.64) | 0.006[b] | 0.08 (0–15.64) | 0.04 (0–15.64) | 0.05 (0–15.64) | 0.652[b] |
| | Mean | 1.04±2.03 | 1.40±2.56 | 1.22±2.32 | | 1.19±2.28 | 1.25±2.36 | 1.22±2.32 | |
| **Site of metastasis** | | | | | | | | | |
| | Bone | 543 (84.8%) | 542 (84.7%) | 1085 (84.8%) | 0.943 | 524 (81.9%) | 561 (87.7%) | 1085 (84.8%) | 0.007 |
| | Brain | 17 (2.7%) | 19 (3.0%) | 36 (2.8%) | | 25 (3.9%) | 11 (1.7%) | 36 (2.8%) | |
| | Others | 80 (12.5%) | 79 (12.3%) | 159 (12.4%) | | 91 (14.2%) | 68 (10.6%) | 159 (12.4%) | |
| **Multiple metastases** | | | | | | | | | |
| | No | 121 (18.9%) | 124 (19.4%) | 245 (19.1%) | 0.832[a] | 120 (18.8%) | 125 (19.5%) | 245 (19.1%) | 0.776[a] |
| | Yes | 519 (81.1%) | 516 (80.6%) | 1035 (80.9%) | | 520 (81.3%) | 515 (80.5%) | 1035 (80.9%) | |
| **Site of primary cancer** | | | | | | | | | |
| | Lung cancer | 253 (39.5%) | 222 (34.7%) | 475 (37.1%) | <0.001[a] | 232 (36.3%) | 243 (38.0%) | 475 (37.1%) | <0.001[a] |
| | Hepatoma | 75 (11.7%) | 60 (9.4%) | 135 (10.5%) | | 75 (11.7%) | 60 (9.4%) | 135 (10.5%) | |
| | Breast cancer | 26 (4.1%) | 90 (14.1%) | 116 (9.1%) | | 56 (8.8%) | 60 (9.4%) | 116 (9.1%) | |

*(Continued)*

**Table 1.** (Continued)

| | | SATI | | | | VATI | | | |
|---|---|---|---|---|---|---|---|---|---|
| | | Low | High | Total number | P value | Low | High | Total number | P value |
| | Prostate cancer | 39 (6.1%) | 53 (8.3%) | 92 (7.2%) | | 21 (3.3%) | 71 (11.1%) | 92 (7.2%) | |
| | Rectal cancer | 34 (5.3%) | 43 (6.7%) | 77 (6.0%) | | 40 (6.3%) | 37 (5.8%) | 77 (6.0%) | |
| | Others | 213 (33.3%) | 172 (26.9%) | 385 (30.1%) | | 216 (33.8%) | 169 (26.4%) | 385 (30.1%) | |
| **EQD$_{2Gy}$** | | | | | | | | | |
| | <32.5 | 337 (52.7%) | 279 (43.6%) | 616 (48.1%) | 0.001[a] | 325 (50.8%) | 291 (45.5%) | 616 (48.1%) | 0.065[a] |
| | ≥32.5 | 303 (47.3%) | 361 (56.4%) | 664 (51.9%) | | 315 (49.2%) | 349 (54.5%) | 664 (51.9%) | |
| | Median | 31.25 (1.44–70.00) | 32.50 (3.25–84.00) | 32.50 (1.44–84.00) | <0.001[b] | 31.98 (1.83–70.00) | 32.50 (1.44–84.00) | 32.50 (1.44–84.00) | 0.484[b] |
| | Mean | 28.14±10.99 | 30.66±10.70 | 29.40±10.91 | | 29.19±11.29 | 29.62±10.52 | 29.40±10.91 | |
| **Systemic therapy** | | | | | | | | | |
| | No | 266 (41.6%) | 164 (25.6%) | 430 (33.6%) | <0.001[a] | 238 (37.2%) | 192 (30.0%) | 430 (33.6%) | 0.008[a] |
| | Yes | 374 (58.4%) | 476 (74.4%) | 850 (66.4%) | | 402 (62.8%) | 448 (70.0%) | 850 (66.4%) | |
| **Comorbidities** | | | | | | | | | |
| | No | 296 (46.3%) | 256 (40.0%) | 552 (43.1%) | 0.028[a] | 328 (51.2%) | 224 (35.0%) | 552 (43.1%) | <0.001[a] |
| | Yes | 344 (53.8%) | 384 (60.0%) | 728 (56.9%) | | 312 (48.8%) | 416 (65.0%) | 728 (56.9%) | |
| **Employment status** | | | | | | | | | |
| | High | 152 (23.8%) | 142 (22.2%) | 294 (23.0%) | <0.001[a] | 152 (23.8%) | 142 (22.2%) | 294 (23.0%) | 0.152[a] |
| | Low | 275 (43.0%) | 169 (26.4%) | 444 (34.7%) | | 234 (36.6%) | 210 (32.8%) | 444 (34.7%) | |
| | None | 213 (33.3%) | 329 (51.4%) | 542 (42.3%) | | 254 (39.7%) | 288 (45.0%) | 542 (42.3%) | |
| **Education level** | | | | | | | | | |
| | None/primary school | 311 (48.6%) | 344 (53.8%) | 655 (51.2%) | 0.074[a] | 291 (45.5%) | 364 (56.9%) | 655 (51.2%) | <0.001[a] |
| | High school | 329 (51.4%) | 296 (46.3%) | 625 (48.8%) | | 349 (54.5%) | 276 (43.1%) | 625 (48.8%) | |
| **Place of residence** | | | | | | | | | |
| | Urban | 351 (54.8%) | 345 (53.9%) | 696 (54.4%) | 0.779[a] | 364 (56.9%) | 332 (51.9%) | 696 (54.4%) | 0.082[a] |
| | Rural | 289 (45.2%) | 295 (46.1%) | 584 (45.6%) | | 276 (43.1%) | 308 (48.1%) | 584 (45.6%) | |
| **Cigarette smoking** | | | | | | | | | |
| | No | 270 (42.2%) | 455 (71.1%) | 725(56.6%) | <0.001[a] | 356 (55.6%) | 369 (57.7%) | 725 (56.6%) | 0.499[a] |
| | Yes | 370 (57.8%) | 185(28.9%) | 555 (43.4%) | | 284 (44.4%) | 271 (42.3%) | 555 (43.4%) | |
| **Betel quid chewing** | | | | | | | | | |
| | No | 511 (79.8%) | 578 (90.3%) | 1089 (85.1%) | <0.001[a] | 538 (84.1%) | 551 (86.1%) | 1089 (85.1%) | 0.347[a] |
| | Yes | 129 (20.2%) | 62 (9.7%) | 191 (14.9%) | | 102 (15.9%) | 89 (13.9%) | 191 (14.9%) | |
| **Alcohol drinking** | | | | | | | | | |
| | No | 420 (65.6%) | 531 (83.0%) | 951 (74.3%) | <0.001[a] | 468 (73.1%) | 483 (75.5%) | 951 (74.3%) | 0.338[a] |
| | Yes | 220 (34.4%) | 109 (17.0%) | 329 (25.7%) | | 172 (26.9%) | 157 (24.5%) | 329 (25.7%) | |
| **Days of metastases treatment** | | | | | | | | | |
| | ≤12 | 360 (56.3%) | 311 (48.6%) | 671 (52.4%) | 0.007[a] | 328 (51.2%) | 343 (53.6%) | 671 (52.4%) | 0.433[b] |
| | ≥13 | 280 (43.8%) | 329 (51.4%) | 609 (47.6%) | | 312 (48.8%) | 297 (46.4%) | 609 (47.6%) | |
| | Median | 11.50 (1–93) | 13.00(1–67) | 12.00(1–93) | <0.001[b] | 12.00 (1–93) | 12.00(1–67) | 12.00(1–93) | 0.984[b] |
| | Mean | 11.60±7.98 | 13.59 ±8.86 | 12.59±8.49 | | 12.59±8.91 | 12.60±8.05 | 12.59±8.49 | |
| **Metastasis treatment period** | | | | | | | | | |
| | ≤2009 | 331 (51.7%) | 298 (46.6%) | 629 (49.1%) | 0.074[a] | 312 (48.8%) | 317 (49.5%) | 629 (49.1%) | 0.823[a] |
| | ≥2010 | 309 (48.3%) | 342 (53.4%) | 651 (50.9%) | | 328 (51.2%) | 323 (50.5) | 651 (50.9%) | |
| **Body mass index, kg/m$^2$** | | | | | | | | | |
| | Underweight | 116 (18.1%) | 4 (0.6%) | 120 (9.4%) | <0.001[a] | 118 (18.4%) | 2 (0.3%) | 120 (9.4%) | <0.001[a] |
| | Normal weight | 478 (74.7%) | 344 (53.8%) | 822 (64.2%) | | 465 (72.7%) | 357 (55.8%) | 822 (64.2%) | |

(*Continued*)

**Table 1.** (Continued)

| | SATI | | | | VATI | | | |
|---|---|---|---|---|---|---|---|---|
| | Low | High | Total number | P value | Low | High | Total number | P value |
| Overweight | 45 (7.0%) | 242 (37.8%) | 287 (22.4%) | | 55 (8.6%) | 232 (36.3%) | 287 (22.4%) | |
| Obese | 1 (0.2%) | 50 (7.9%) | 51 (4.0%) | | 2 (0.3%) | 49 (7.7%) | 51 (4.0%) | |
| Median | 21.01 (13.34–20.65) | 24.78 (16.98–38.73) | 22.86 (13.34–38.73) | <0.001[b] | 21.01 (13.34–30.65) | 24.68 (16.98–38.73) | 22.86 (13.34–38.73) | <0.001[b] |
| Mean | 21.12±2.74 | 25.08±3.32 | 23.10±3.63 | | 21.12±2.74 | 25.08±3.32 | 23.10± 3.63 | |

Abbreviations: SD, standard deviation; ECOG, Eastern Cooperative Oncology Group; EQD2 Gy, equivalent doses in 2-Gy fractions; SATI, subcutaneous adipose tissue index; VATI, visceral adipose tissue index.

[a]Chi-square test

[b]ANOVA test

date last known to be alive. The study cohort included 740 (57.8%) men and 540 (42.2%) women. The most common primary cancer site was the lung (35% in both sexes), and there were 897 (70%) patients with an ECOG performance status of 0−1. The interval between the diagnosis of primary cancer and the detection of metastases was 0.11 months in women (95% CI: 0−15.64 months) and 0.04 months (95% CI: 0−13.50 months) in men, respectively. Table 1 shows the results pertaining to adiposity indices. Men had higher skeletal muscle and VATI than women, whereas SATI was higher in women.

## Survival analysis

The median follow-up time for the 43 surviving patients was 78.28 months (range: 0.789−147.25 months). The median OS after RT was 6.03 months (range: 0.03−147.25 months). The 6-, 12-, 24-, and 48-month OS rates in women and men were 41.4%/61.8%, 23.6% /43.2%, 9.8%/22.6%, and 3.8%/11.2%, respectively. The median OS was 9.53 months (range: 0.10−137.42 months) in women and 4.7 months (range: 0.30−147.25) in men.

SATI values $\geq$11.63 cm$^2$ in men and $\geq$25.21 cm$^2$ in women were considered as high. Similarly, VATI values $\geq$10.46 cm$^2$ in men and $\geq$8.96 cm$^2$ in women were regarded as elevated. The median OS in the high and low SATI groups was 27.77 months (range: 15.50−148.77 months) and 8.15 months (range: 0.01−15.48 months), respectively. The median OS in the high and low VATI groups was 19.35 months (range: 9.84−93.34 months) and 3.70 months (range: 0.02−9.83 months), respectively.

The results of univariate and multivariate analyses are presented in Table 2. The following variables were independently associated with OS in multivariate analysis: SATI, VATI, sex, performance status, primary tumor site, more than one metastatic site, ECOG performance status, EQD2Gy, systemic therapy, education, days of metastases treatment, and time to metastases (Table 2).

## Prognostic significance of SATI and VATI

We subsequently examined the prognostic impact of SATI and VATI by classifying patients into high *versus* low categories. Kaplan-Meier analysis revealed no differences in OS between the high SATI/high VATI group (median survival: 9.37 months) and high SATI/low VATI group (median survival: 9.43 months; P = 0.303; Table 3). The lowest OS (3.97 months) was observed in the low SATI/low VATI group (Fig 1; Table 3).

**Table 2. Univariate and multivariate analysis of overall survival.**

| Number of patients = 1280 | Univariate analysis Overall survival HR (95% CI) | P value | Multivariable analysis Overall survival HR (95% CI) | P value |
|---|---|---|---|---|
| SATI (high *versus* low) | 0.551 (0.492–0.618) | <0.001 | 0.696 (0.606–0.800) | <0.001 |
| VATI (high *versus* low) | 0.756 (0.676–0.846) | <0.001 | 0.870 (0.764–0.992) | 0.037 |
| Muscle index (high *versus* low) | 1.042 (0.932–1.165) | 0.470 | | |
| Age group ($\geq$59.5 *versus* <59.5 years) | 1.186 (1.061–1.327) | 0.003 | 0.972 (0.848–1.113) | 0.679 |
| Sex (male *versus* female) | 1.579 (1.408–1.770) | <0.001 | 1.186 (1.010–1.393) | 0.037 |
| ECOG performance status (2–4 *versus* 0–1) | 1.257 (1.113–1.420) | <0.001 | 1.305 (1.153–1.478) | <0.001 |
| Multiple metastases (yes *versus* no) | 1.414 (1.223–1.635) | <0.001 | 1.350 (1.165–1.565) | <0.001 |
| Site of primary cancer (lung *versus* other sites) | 0.816 (0.727–0.916) | 0.001 | 0.813 (0.716–0.922) | 0.001 |
| EQD$_{2Gy}$ ($\geq$32.5 *versus* <32.5) | 0.651 (0.582–0.729) | <0.001 | 0.802 (0.695–0.925) | 0.002 |
| Systemic therapy (yes *versus* no) | 0.584 (0.519–0.658) | <0.001 | 0.621 (0.546–0.706) | <0.001 |
| Comorbidities (yes *versus* no) | 1.120 (1.000–1.253) | 0.049 | 1.065 (0.948–1.198) | 0.288 |
| Education level (high school *versus* none/primary school) | 0.864 (0.772–0.966) | 0.010 | 0.869 (0.763–0.989) | 0.033 |
| Cigarette smoking (yes *versus* no) | 1.432 (1.279–1.603) | <0.001 | 1.092 (0.934–1.276) | 0.272 |
| Betel quid chewing (yes *versus* no) | 1.331 (1.139–1.555) | <0.001 | 0.964 (0.804–1.155) | 0.689 |
| Alcohol drinking (yes *versus* no) | 1.373 (1.209–1.560) | <0.001 | 1.148 (0.989–1.332) | 0.069 |
| Days of metastasis treatment ($\geq$13 *versus* $\leq$12) | 0.704 (0.629–0.787) | <0.001 | 0.864 (0.750–0.996) | 0.044 |
| Onset of metastasis (>1 year *versus* $\leq$1 year) | 0.800 (0.708–0.904) | <0.001 | 0.913 (0.801–1.041) | 0.175 |
| Place of residence (urban *versus* rural) | 0.985 (0.881–1.102) | 0.795 | | |
| Site of metastasis | | 0.216 | | |
| Bone | 1.340 (0.957–1.877) | 0.089 | | |
| Brain | 0.974 (0.821–1.156) | 0.765 | | |
| Employment status | | 0.556 | | |
| (low *versus* high) | 1.084 (0.932–1.260) | 0.294 | | |
| (none *versus* high) | 1.034 (0.895–1.195) | 0.652 | | |
| Metastases treatment period ($\geq$2010 *versus* $\leq$2009) | 0.990 (0.886–1.107) | 0.866 | | |
| Body mass index (>25 *versus* $\leq$25 kg/m$^2$) | 0.755 (0.664–0.857) | <0.001 | | |

Abbreviations: HR, hazard ratio; CI, confidence interval; ECOG, Eastern Cooperative Oncology Group; EQD2 Gy, equivalent doses in 2-Gy fractions; SATI, subcutaneous adipose tissue index; VATI, visceral adipose tissue index. An L3 subcutaneous adipose tissue index $\geq$11.63 cm$^2$ m$^{-2}$ in males and $\geq$25.21 cm$^2$ m$^{-2}$ in females was considered as high. An L3 visceral adipose tissue index $\geq$10.46 cm$^2$ m$^{-2}$ in males and $\geq$8.96 cm$^2$ m$^{-2}$ in females was considered as high.

## Prognostic stratification according to sex and body composition

Thereafter, both sex and SATI values were taken into account to construct four different groups. We specifically selected SATI owing to its higher prognostic value in multivariate analysis. A total of four groups were identified (male/high SATI; female/high SATI; male/low SATI; female/low SATI), with the most favorable survival figures being evident in the female/high SATI group (median OS: 11.21 months; 95% CI: 9.434–12.988 months; P<0.001 *versus* other groups). The less favorable OS survival (median: 3.847 months; 95% CI: 3.391–4.302 months) was observed in the male/low SATI group (Fig 2; Table 4).

**Table 3. Multivariate Cox regression analysis of overall survival in patients stratified according to subcutaneous adiposity and visceral adiposity.**

| Subgroup | No. of patients | Median survival time (95% CI) | HR | P value |
|---|---|---|---|---|
| High SATI/high VATI | 455 | 9.370 (8.116–10.624) | | <0.001[a] |
| High SATI/low VATI | 185 | 9.436 (7.129–11.742) | 1.097 (0.920–1.307) | 0.303 |
| Low SATI/high VATI | 185 | 4.603 (3.657–5.548) | 1.882 (1.580–2.242) | <0.001[a] |
| Low SATI/low VATI | 455 | 3.978 (3.362–4.594) | 1.854 (1.622–2.121) | <0.001[a] |

Abbreviations: CI, confidence interval; HR, hazard ratio; SATI, subcutaneous adipose tissue index; VATI, visceral adipose tissue index.

[a]Chi-square test

[b]ANOVA test.

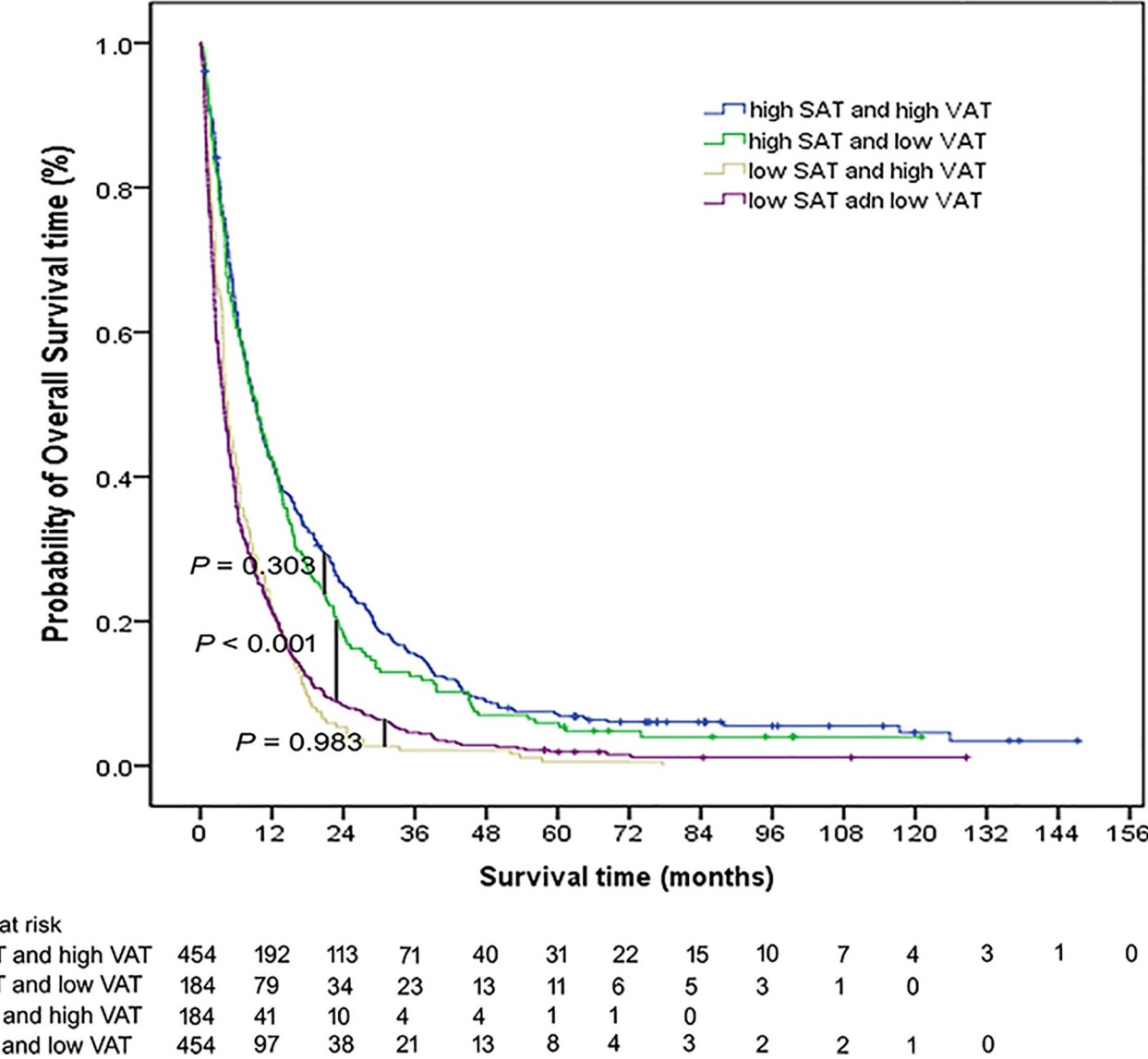

| No. at risk | | | | | | | | | | | | | |
|---|---|---|---|---|---|---|---|---|---|---|---|---|---|
| high SAT and high VAT | 454 | 192 | 113 | 71 | 40 | 31 | 22 | 15 | 10 | 7 | 4 | 3 | 1 | 0 |
| high SAT and low VAT | 184 | 79 | 34 | 23 | 13 | 11 | 6 | 5 | 3 | 1 | 0 |
| low SAT and high VAT | 184 | 41 | 10 | 4 | 4 | 1 | 1 | 0 |
| low SAT and low VAT | 454 | 97 | 38 | 21 | 13 | 8 | 4 | 3 | 2 | 2 | 1 | 0 |

**Fig 1. Kaplan-Meier estimates of overall survival in patients with bone metastases stratified according to subcutaneous adiposity (high *versus* low) and visceral adiposity (high *versus* low).**

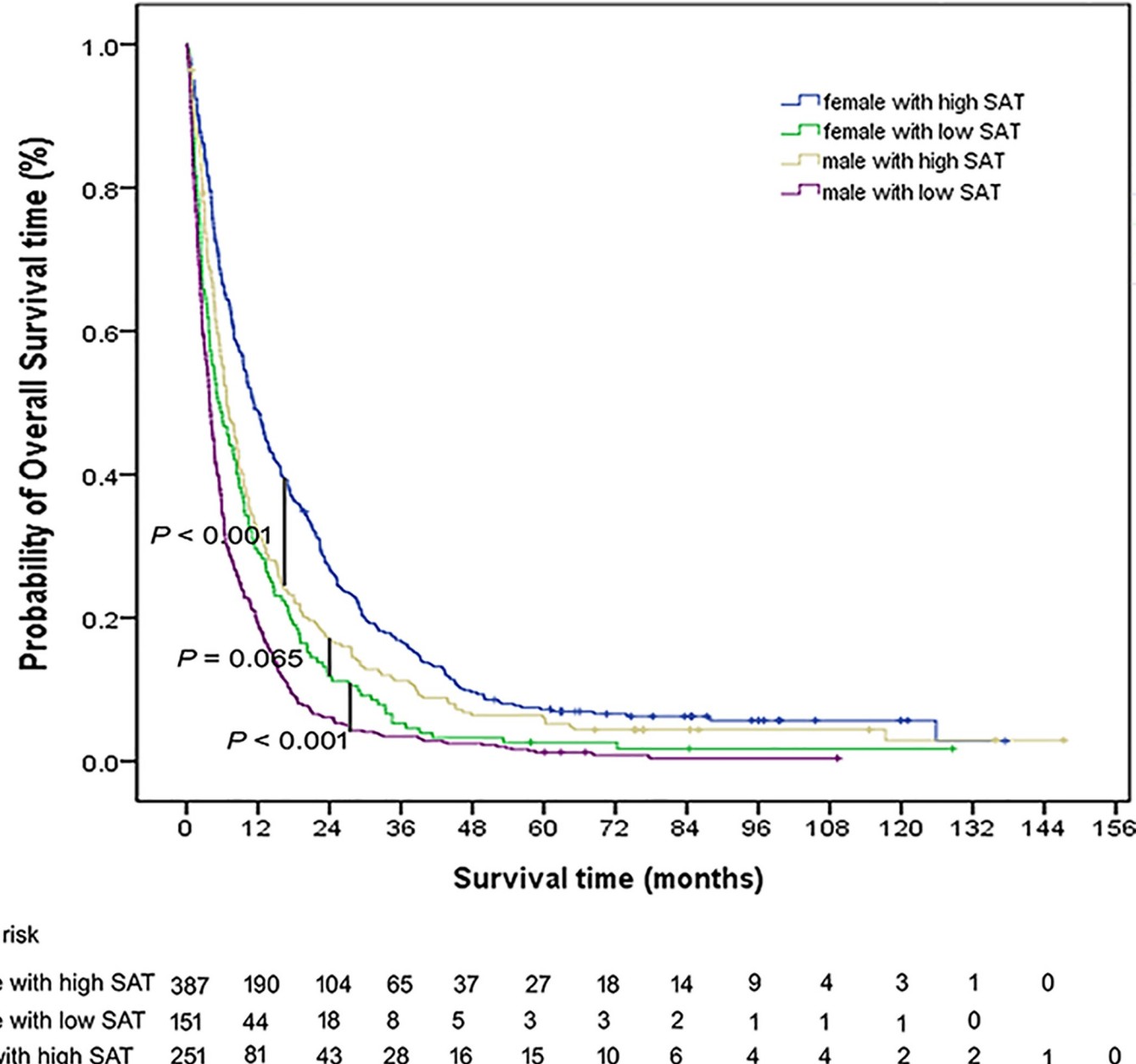

**Fig 2. Kaplan-Meier estimates of overall survival in patients with bone metastases stratified according to subcutaneous adiposity (high *versus* low) and sex (female *versus* male).**

## Discussion

The results of this retrospective analysis of prospectively collected data can be summarized as follows: 1) high SATI and VATI were independently associated with a better OS in a sample of Taiwanese patients with bone metastases, with the former showing a stronger relationship; 2) the most favorable OS was observed in women with high SATI. Although we observed associations–and not prediction or causation–our study adds to the growing literature investigating adiposity in relation to clinical outcomes of patients with malignancies.[13–15]

**Table 4. Multivariate Cox regression analysis of overall survival in patients stratified according to sex and subcutaneous adiposity.**

| Subgroup | No. of patients | Median survival time (95%CI) | HR | P value |
|---|---|---|---|---|
| Female with high SATI | 388 | 11.211 (9.434–12.988) | | <0.001[a] |
| Female with low SATI | 152 | 5.293 (3.084–7.503) | 1.604 (1.324–1.942) | <0.001[a] |
| Male with high SATI | 252 | 6.773 (5.481–8.064) | 1.312 (1.115–1.545) | <0.001[a] |
| Male with low SATI | 488 | 3.847 (3.391–4.302) | 2.182 (1.899–2.507) | <0.001[a] |

Abbreviations: CI, confidence interval; HR, hazard ratio; SATI, subcutaneous adipose tissue index.

[a]Chi-square test

[b]ANOVA

Currently, the association of indices of adiposity with the disease course of cancer patients remains controversial. Although adiposity seems to be positively correlated with OS in several solid tumors [16, 17], poorer survival figures have been reported for obese patients with cancer–possibly because of an increased production of growth factors and inflammatory mediators from the adipose tissue.[18] In this regard, it should be noted that adipose tissue may serve as a nutrient replacement in patients with cancer, [15, 19] but it can be also involved in tumor spread through adipokine-induced extracellular matrix remodeling.[20]

Pai et al. [21] have previously shown that SATI is strongly related to distant metastasis-free survival, locoregional control, and OS in 881 patients with head and neck cancer. Ebadi et al. [5] also demonstrated that patients with low SATI and high VATI independently predicted mortality in a sample of patients with different solid malignancies. Herein, we show that VATI, and most prominently SATI, were significantly associated with OS in Taiwanese patients with bone metastases. Controversy still exists on the relationship between VATI and clinical outcomes in patients with solid tumors.[10, 22–24] The reasons whereby SATI appears to hold a stronger association with OS over VATI in our study remain to be elucidated. However, it is notable that–differently from visceral fat (which is an active endocrine organ)–subcutaneous fat is more strictly involved in lipid and energy storage and is characterized by a lower inflammatory environment [14, 25, 26].

The study was conducted in a radiation oncology setting. Bone metastases are not only the most common site of distant spread in patients with solid malignancies but they are also the most commonly identified by radiation oncologists. The question as to whether our findings may be applied to patients with metastases to other distant sites (e.g., liver or brain) remains open. We acknowledge several limitations of the present study. First, the study was conducted in an Asian population, and it is well-known that ethnic differences exist in measures of adiposity between Asian and Caucasian populations [27]. Therefore, our findings need to be independently replicated in other geographic areas. Second, we did not segment body fat in the whole CT volume. Nonetheless, there is published evidence suggesting that measures of adiposity obtained at the L3 level through a simplified CT protocol are well-correlated to those taken at other sites [28–31]. Third, all measures of adiposity were taken in the two week preceding the start of RT. Wu et al. [32] have recently demonstrated the prognostic importance of the time at which body adiposity is assessed. However, these data were not available in this study, and we were unable to run this analysis. Finally, this was a retrospective analysis of prospectively collected data which had an exploratory nature. The application of the Bonferroni's correction in this setting may be too conservative and was avoided. In any case, our results should be considered as preliminary and hypothesis-generating. Because we observed associations, we cannot claim any prognostic effect of adiposity indices in our population. Future longitudinal studies are required to clarify this issue further.

These limitations notwithstanding, we found that high SATI and VATI are associated with a more favorable OS in Taiwanese patients with bone metastases referred for RT. The question as to whether clinical measures aimed at improving adiposity may improve OS in this clinical population deserves further scrutiny.

## Acknowledgments

The authors thank all the patients for the participation in this study.

## Author Contributions

**Conceptualization:** Chi Cheng Chuang, Kai Ping Chang, Ping Ching Pai, Kuan Hung Chen, Wen Chi Chou, Shiao Fwu Tai.

**Data curation:** Ngan Ming Tsang, Kin Fong Lei.

**Formal analysis:** Wen Ching Chuang, Ngan Ming Tsang.

**Investigation:** Wen Ching Chuang.

**Methodology:** Shu Chen Liu.

**Resources:** Chi Cheng Chuang, Kai Ping Chang, Wen Chi Chou.

**Supervision:** Ngan Ming Tsang.

**Writing – original draft:** Wen Ching Chuang.

**Writing – review & editing:** Ngan Ming Tsang.

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
