## [Decision Letter · Decision Letter 0]

2 Dec 2019

PONE-D-19-27964

Independent prognostic significance of subcutaneous and visceral adipose tissue in patients with bone metastases

PLOS ONE

Dear Dr. Tsang,

Thank you for submitting your manuscript to PLOS ONE. After careful consideration, we feel that it has merit but does not fully meet PLOS ONE’s publication criteria as it currently stands. Therefore, we invite you to submit a revised version of the manuscript that addresses the points raised during the review process.

We would appreciate receiving your revised manuscript by Jan 16 2020 11:59PM. To enhance the reproducibility of your results, we recommend that if applicable you deposit your laboratory protocols in protocols.io, where a protocol can be assigned its own identifier (DOI) such that it can be cited independently in the future. For instructions see: http://journals.plos.org/plosone/s/submission-guidelines#loc-laboratory-protocols

We look forward to receiving your revised manuscript.

Kind regards,

Mauro Lombardo

Academic Editor

PLOS ONE

Journal Requirements:

2.  Thank you for including your ethics statement; " The Institutional Review Broad (IRB) approval number was IRB201701224B0."

3. In ethics statement in the manuscript and in the online submission form, please provide additional information about the patient records/samples used in your retrospective study. Specifically, please ensure that you have discussed whether all data/samples were fully anonymized before you accessed them and/or whether the IRB or ethics committee waived the requirement for informed consent. If patients provided informed written consent to have data/samples from their medical records used in research, please include this information.

Additional Editor Comments (if provided):

Please follow point by point reviewers' comments before resubmitting.

Reviewers' comments:

Reviewer's Responses to Questions

**Comments to the Author**

1. Is the manuscript technically sound, and do the data support the conclusions?

Reviewer #1: Partly

Reviewer #2: Yes

Reviewer #3: Partly

2. Has the statistical analysis been performed appropriately and rigorously? 

Reviewer #1: No

Reviewer #2: Yes

Reviewer #3: Yes

3. Have the authors made all data underlying the findings in their manuscript fully available?

Reviewer #1: No

Reviewer #2: No

Reviewer #3: Yes

4. Is the manuscript presented in an intelligible fashion and written in standard English?

Reviewer #1: Yes

Reviewer #2: Yes

Reviewer #3: Yes

5. Review Comments to the Author

Reviewer #1: In the present study, the authors retrospectively investigate the association between SATI, VATI and survival in patients with bone metastases. Although the topic is interesting and the manuscript is overall well written, there are several issues as follows:

1. The association between body fat and survival in many cancers is well-known and established. Hence, the study does not add much new to the current knowledge. However, I acknowledge that this is not necessarily relevant for PLoS ONE, so this point may be disregarded by the editor.

2. The study rationale is not quite clear. Why focus on patients with bone metastases? Other metastases also herald poor prognosis (liver, brain ...).

3. The chance was missed to segment the body fat in the whole CT scan volume and thus provide a more accurate estimate.

4. A biostatistician should be consulted for the statistical analysis, which is not well done (the current analysis leads to spuriously low p-values).

5. The term "predictor" should be avoided throughout the manuscript. In a retrospective study, one can only find associations. Predictions can only be made with a prospective trial. This leads me to the next point:

6. The authors grossly overinterpret the results of their study. Although this is unfortunately the norm in many scientific journals where authors need to "sell" their study, per PLoS ONE policy this is not needed. To give the authors an example: "More importantly, our findings may pave the way for aggressive therapeutic interventions in the subset of patients who are expected to have more favorable survival figures" To make such a statement, a prospective randomised clinical trial is needed.

Reviewer #2: 1. Summary of the research and overall impression

The research entitled "Independent prognostic significance of subcutaneous and visceral adipose tissue in patients with bone metastases" is an interesting and pertinent topic in current discussions of the academic literature.

The main research question addressed by the authors is to discuss whether regional adiposity measures such as subcutaneous adipose tissue index (SATI) and visceral adipose tissue index (VATI) can predict overall survival (OS) in cancer patients with bone metastases.

There is a common perception that, compared with normal-weight patients, elevated BMI is also associated with poorer prognosis after cancer diagnosis. However, some studies have challenged this thinking by showing that among cancer patients, high BMI is associated with improved survival compared with normal weight patients. This finding is known as the obesity paradox, well established in the cardio-metabolic literature, but less appreciated in oncology.

However, some observations of the obesity paradox in cancer reflect methodological mechanisms including the crudeness of BMI as an obesity measure, confounding, detection bias, reverse causality, and selection bias. Thus, when we come across studies on this subject, one should pay attention to these questions so as not to incur mistaken associations.

An important issue to note is that BMI is a relatively crude measure of body adiposity and body composition and does not differentiate between lean mass and fat mass. In turn, body composition varies with age, sex, and ethnicity, such that there are currently no specific age-gender-ethnicity indices to define obesity in a standardized manner. Thus, for example, in a cancer population, overweight individuals (defined by BMI) might be younger with high muscle mass (compared with normal weight), explaining their better outcome compared with normal weight.

The paradox might not exist if alternate measures of body composition or adipose tissue were used. Thus, for example, we found no examples of studies in patients with cancer demonstrating the obesity paradox when anthropometric measures other than BMI or body composition indices were used.

We note that in this study the use of these alternative measures of body composition was used. The authors used regional measures of adiposity, such as the subcutaneous adipose tissue index (SATI) and the visceral adipose tissue index (VATI), which reduces the risk of this paradox and strengthens the consistency of the results presented. We consider this a strength of this study.

The manuscript is technically adequate to its purpose. In analyzing the data provided by the authors, we point out that the results support the conclusions. The information contained in table 1 (p.11-15), table 2 (p.17-19), table 3 (p.21), table 4 (p.23), fig 1 (p.20) and fig 2 (p.22), are well described and provide the necessary information to support the conclusions of this study. The information described in the materials and methods section is clearly written and allows for reproducibility of the study. We also emphasize that the sample size was expressive (1280 cases) and thus appropriate.

However, it is important to emphasize that the use of retrospective design used in this study favors the selection bias and does not allow mechanistic conclusions about the associations found by the authors, this being a weakness of this study. However, they contribute to point out some important paths to follow in future research.

The strategies used to perform the statistical analysis are consistent and suitable for the treatment of the collected data. It is noteworthy that the choice of variables included in the univariate analysis of overall survival, such as cigarette smokin, alcohol drinking, multiple metastases, among others, were important to minimize confounding factors.

When searching this manuscript in several academic databases, no duplicity was found. Therefore, it is believed that the reported results were not published anywhere else.

The research meets the standards applicable to the ethics of experimentation and the integrity of research in humans and has been approved by the Institutional Review Broad Research Ethics Committee (approval number IRB201701224B0).

The authors did not use any public repository to provide data from this study. However, all relevant data are within the manuscript and its supporting information files.

After reviewing this manuscript, considering the editorial criteria for publication employed by PLOS ONE, I suggest approval for publication with “Minor Revision”.

2. Discussion of specific areas for improvement

Major issues

No major issues were identified in the manuscript that compromise the results observed by the authors.

Minor issues

Technical clarifications:

On page 25, the authors report as a limitation of this study: "Body composition was assessed in all patients prior to the first RT session and not when bone metastases were diagnosed." However, they do not clearly explain how this may imply the results found. Thus, I suggest to the authors, as a minor review, to further clarify this issue, given that the time when BMI was determined is relevant to the observed patterns of association.

We suggest the authors review the World Cancer Research Fund (WCRF) report on the effect of risk factors on survival among women with breast cancer. This recent report added a very useful classification—namely, determination of BMI either at pre-, peri-, or post-diagnosis (the later typically 12 months after the initial treatment) of câncer. From these, different patterns of associations emerge. In a meta-analysis of 29 studies evaluating the impact of BMI on survival in patients with colorectal cancer, Wu et al. observed that increasing pre-diagnosis BMI prognosticated for a poor survival but that post-treatment overweight was associated with improved survival, i.e., the obesity paradox.

Suggested References to Authors:

WCRF. World Cancer Research Fund International. Continuous Update Project Report: diet, nutrition, physical activity, and breast cancer survivors. 2014. Available at:www.wcrf.org/sites/default/files/Breast-Cancer-Survivors-2014-Report.pdf [Accessed 20th Dec 2014]. 2014.

Wu S, Liu J, Wang X, Li M, Gan Y, Tang Y. Association of obesity and overweight with overall survival in colorectal cancer patients: a meta-analysis of 29 studies. Cancer Causes Control. 2014;25:1489–1502. doi: 10.1007/s10552-014-0450-y.

Reviewer #3: I consider the piece is valuable. However, results must be explained more in detail to be able to trace interpretations offered at discussion. Manuscript should be reviewed for grammatical errors (for example patient demographics, should be more in relation to the concept assessment than the way variables were implemented), specially for critical interpretations of data and analyses. Recommendations for more aggressive therapeutics is not included in the analysis, but is being offered as a conclusion, therefore I recommend to review this issue and explain more about how this is derived from the analysis.

6. PLOS authors have the option to publish the peer review history of their article (what does this mean?). If published, this will include your full peer review and any attached files.

Reviewer #1: No

Reviewer #2: No

Reviewer #3: Yes: Milena Castro

---

## [Author Response · Author response to Decision Letter 0]

20 Dec 2019

AUTHORS’ REPLY TO REVIEWER’S #1 COMMENTS

1. The association between body fat and survival in many cancers is well-known and established. Hence, the study does not add much new to the current knowledge. However, I acknowledge that this is not necessarily relevant for PLoS ONE, so this point may be disregarded by the editor.

REPLY. We thank the Reviewer for the constructive criticism. We concur that the association between adiposity and survival in patients with malignancies has been extensively investigated in the past. However, this study may represent a valid addition to the existing literature for the following reasons: 1) the association of subcutaneous adiposity and visceral adiposity with survival endpoints in patients with cancer remains controversial, 2) we specifically focused on patients with bone metastases who were referred for radiotherapy (see below for further clarifications on study population); 3) the study was conducted in an Asian population (it is well-known that ethnic differences exist in measures of adiposity between Asian and Caucasian populations). We believe that these points should be considered when dealing with the novelty of our work. Please note that the title has been revised to highlight the ethnic origin of the study population.

The study rationale is not quite clear. Why focus on patients with bone metastases? Other metastases also herald poor prognosis (liver, brain ...).

REPLY. We are grateful to the Reviewer for the cogent comment. We specifically focused on bone metastases for the following reasons. First, the study was conducted in a radiation oncology setting. Bone metastases are not only the most common site of distant spread in patients with solid malignancies but they are also the most commonly identified by radiation oncologists. We did focus on bone metastases not only because they portend a poor prognosis, but also in light of their high frequency. Second, patients with bone metastases commonly present with pain and an impaired quality of life, ultimately requiring referral for radiotherapy. We therefore believe that our study population is of special interest for radiation therapists. We nonetheless acknowledged the lack of inclusion of patients with metastases at other sites as a limitation inherent in our study (please see the revised “Discussion” section). 

The chance was missed to segment the body fat in the whole CT scan volume and thus provide a more accurate estimate.

REPLY. We concur with the Reviewer that whole-body computed tomography (CT) can actually provide a more accurate and comprehensive estimation of adiposity. However, there is published evidence (see references 1−4 below) suggesting that measures of adiposity obtained at the L3 level through a simplified CT protocol are well-correlated to those taken at other sites. We are aware that this is a potential limitation inherent in our study, which has been acknowledged in the revised “Discussion” section. 

A biostatistician should be consulted for the statistical analysis, which is not well done (the current analysis leads to spuriously low p-values).

REPLY. This study was designed as a retrospective analysis of prospectively collected data (a point which has been clarified in revised paper). The statistical approach was in line with the analyses conducted in other published papers from our group (see references 5−8 below). With regard to the low p-values, they may stem from a multiple comparison problem. However, the application of the Bonferroni’s correction may be too conservative in an exploratory analysis. We nonetheless believe that this is a potential caveat inherent in our study, which has been addressed in the revised “Discussion” section. 

5. The term "predictor" should be avoided throughout the manuscript. In a retrospective study, one can only find associations. Predictions can only be made with a prospective trial. This leads me to the next point

REPLY. We thank the Reviewer for the pertinent observation. In the revised version of the paper, we made appropriate revisions to highlight that we observed associations, not prediction or causation. We also acknowledged the retrospective design as a major study limitation in the “Discussion” section.

6. The authors grossly overinterpret the results of their study. Although this is unfortunately the norm in many scientific journals where authors need to "sell" their study, per PLoS ONE policy this is not needed. To give the authors an example: "More importantly, our findings may pave the way for aggressive therapeutic interventions in the subset of patients who are expected to have more favorable survival figures" To make such a statement, a prospective randomised clinical trial is needed.

REPLY. The Reviewer is entirely right and we apologize for the excessive emphasis put on the significance of our findings. In the revised version of the paper, several statements were toned down, the main limitations were highlighted, and the conclusions were drawn more prudently. We have highlighted these points as a relevant future research topic in the revised “Discussion” section.

References

1. Shen W, Punyanitya M, Wang Z, Gallagher D, St-Onge MP, Albu J, Heymsfield SB, Heshka S. Visceral adipose tissue: relations between single-slice areas and total volume. Am J Clin Nutr 2004;80:271–278. 

2. Noumura Y, Kamishima T, Sutherland K, Nishimura H. Visceral adipose tissue area measurement at a single level: can it represent visceral adipose tissue volume? Br J Radiol 2017;90:20170253.

3. Schweitzer L, Geisler C, Pourhassan M, Braun W, Gluer CC, Bosy- Westphal A, Muller MJ. What is the best reference site for a single MRI slice to assess whole-body skeletal muscle and adipose tissue volumes in healthy adults? Am J Clin Nutr 2015;102:58–65. 

4. Demerath EW, Shen W, Lee M, Choh AC, Czerwinski SA, Siervogel RM, Towne B. Approximation of total visceral adipose tissue with a single magnetic resonance image. Am J Clin Nutr 2007;85:362-368.

5. Tsang NM, Pai PC, Chuang CC, Chuang WC, Tseng CK, Chang KP, Yen TC, Lin JD, Chang JT. Overweight and obesity predict better overall survival rates in cancer patients with distant metastases. Cancer Med 2016;5(4):665–675. 

6. Pai PC, Chuang CC, Chuang WC, Tsang NM, Tseng CK, Chen KH, et al. Pretreatment subcutaneous adipose tissue predicts the outcomes of patients with head and neck cancer receiving definitive radiation and chemoradiation in Taiwan. Cancer Med. 2018;7:1630–1641.

7. Pai PC, Chuang CC, Tseng CK, Tsang NM, Chang KP, Yen TC, et al. Impact of pretreatment body mass index on patients with head-and-neck cancer treated with radiation. Int J Radiat Oncol Biol Phys. 2012;83:e93-e100.

8. Tsang NM, Chuang CC, Tseng CK, Hao SP, Kuo TT, Lin CY et al. Presence of the latent membrane protein 1 gene in nasopharyngeal swabs from patients with mucosal recurrent nasopharyngeal carcinoma. Cancer 2003;98:2385–2392.

AUTHORS’ REPLY TO REVIEWER’S #2 COMMENTS

On page 25, the authors report as a limitation of this study: "Body composition was assessed in all patients prior to the first RT session and not when bone metastases were diagnosed." However, they do not clearly explain how this may imply the results found. Thus, I suggest to the authors, as a minor review, to further clarify this issue, given that the time when BMI was determined is relevant to the observed patterns of association.

We suggest the authors review the World Cancer Research Fund (WCRF) report on the effect of risk factors on survival among women with breast cancer. This recent report added a very useful classification—namely, determination of BMI either at pre-, peri-, or post-diagnosis (the later typically 12 months after the initial treatment) of câncer. From these, different patterns of associations emerge. In a meta-analysis of 29 studies evaluating the impact of BMI on survival in patients with colorectal cancer, Wu et al. observed that increasing pre-diagnosis BMI prognosticated for a poor survival but that post-treatment overweight was associated with improved survival, i.e., the obesity paradox.

Suggested References to Authors:

WCRF. World Cancer Research Fund International. Continuous Update Project Report: diet, nutrition, physical activity, and breast cancer survivors. 2014. Available at:www.wcrf.org/sites/default/files/Breast-Cancer-Survivors-2014-Report.pdf [Accessed 20th Dec 2014]. 2014.

Wu S, Liu J, Wang X, Li M, Gan Y, Tang Y. Association of obesity and overweight with overall survival in colorectal cancer patients: a meta-analysis of 29 studies. Cancer Causes Control. 2014;25:1489–1502. doi: 10.1007/s10552-014-0450-y.

REPLY. We thank the Reviewer for the constructive observations. In the revised version of the paper, we clarified that all measures of adiposity were taken in the two week preceding the start of RT. With regard to the World Cancer Research Fund (WCRF) report and the paper by Wu et al., we realize the prognostic importance of the time at which body adiposity is assessed. However, these data were not available in this study, and we were therefore unable to run this analysis. We have highlighted this point as a relevant future research topic in the revised “Discussion” section.

We had mentioned in the Discussion line 256: “Fourth, all measures of adiposity were taken in the two week preceding the start of RT. Wu et al. (32) have recently demonstrated the prognostic importance of the time at which body adiposity is assessed. However, the data of post-diagnosis of cancer (12 months after initial treatment) were not available in this study, and we were unable to run this analysis. ” 

AUTHORS’ REPLY TO REVIEWER’S #3 COMMENTS

I consider the piece is valuable. However, results must be explained more in detail to be able to trace interpretations offered at discussion. Manuscript should be reviewed for grammatical errors (for example patient demographics, should be more in relation to the concept assessment than the way variables were implemented), specially for critical interpretations of data and analyses. Recommendations for more aggressive therapeutics is not included in the analysis, but is being offered as a conclusion, therefore I recommend to review this issue and explain more about how this is derived from the analysis.

REPLY. We thank the Reviewer for the constructive criticisms. In order to address the concern, the following changes were implemented: 1) the paper has been revised for style and presentation; 2) the nature of the analysis was clarified (retrospective analysis of prospectively collected data) and any reference to prediction or causation were removed; 3) we removed the emphasis on therapeutic recommendations because we realize that they are unwarranted based on our current data.

---

## [Decision Letter · Decision Letter 1]

14 Jan 2020

Association of subcutaneous and visceral adipose tissue with overall survival in Taiwanese patients with bone metastases – results from a retrospective analysis of consecutively collected data

PONE-D-19-27964R1

Dear Dr. Tsang,

We are pleased to inform you that your manuscript has been judged scientifically suitable for publication and will be formally accepted for publication once it complies with all outstanding technical requirements.

With kind regards,

Mauro Lombardo

Academic Editor

PLOS ONE

Additional Editor Comments (optional):

Reviewers' comments:

Reviewer's Responses to Questions

**Comments to the Author**

1. If the authors have adequately addressed your comments raised in a previous round of review and you feel that this manuscript is now acceptable for publication, you may indicate that here to bypass the “Comments to the Author” section, enter your conflict of interest statement in the “Confidential to Editor” section, and submit your "Accept" recommendation.

Reviewer #2: All comments have been addressed

Reviewer #3: All comments have been addressed

2. Is the manuscript technically sound, and do the data support the conclusions?

Reviewer #2: Yes

Reviewer #3: Yes

3. Has the statistical analysis been performed appropriately and rigorously? 

Reviewer #2: Yes

Reviewer #3: Yes

4. Have the authors made all data underlying the findings in their manuscript fully available?

Reviewer #2: Yes

Reviewer #3: Yes

5. Is the manuscript presented in an intelligible fashion and written in standard English?

Reviewer #2: Yes

Reviewer #3: Yes

6. Review Comments to the Author

Reviewer #2: We identified from line 247 of the revised “Discussion” section that adjustments have been made to the text to justify the prognostic significance of the time when body adiposity is assessed and highlighted by the authors as a relevant topic for future research. However, the authors adequately responded to the observations and suggestions for improvement we proposed. We agree to approve the revised manuscript for publication.

Reviewer #3: (No Response)

7. PLOS authors have the option to publish the peer review history of their article (what does this mean?). If published, this will include your full peer review and any attached files.

Reviewer #2: No

Reviewer #3: Yes: Milena Castro

---

## [Editor Report · Acceptance letter]

16 Jan 2020

PONE-D-19-27964R1 

Association of subcutaneous and visceral adipose tissue with overall survival in Taiwanese patients with bone metastases – results from a retrospective analysis of consecutively collected data 

Dear Dr. Tsang:

I am pleased to inform you that your manuscript has been deemed suitable for publication in PLOS ONE. Congratulations! Your manuscript is now with our production department. 

With kind regards,

on behalf of

Dr. Mauro Lombardo 

Academic Editor

PLOS ONE